# Atomic scale symmetry and polar nanoclusters in the paraelectric phase of ferroelectric materials

Andreja Bencan[1], Emad Oveisi [2], Sina Hashemizadeh[3,7], Vignaswaran K. Veerapandiyan [4], Takuya Hoshina [5], Tadej Rojac[1], Marco Deluca[4], Goran Drazic [6] & Dragan Damjanovic [3✉]

The nature of the "forbidden" local- and long-range polar order in nominally non-polar paraelectric phases of ferroelectric materials has been an open question since the discovery of ferroelectricity in oxide perovskites, $ABO_3$. A currently considered model suggests locally correlated displacements of B-site atoms along a subset of <111> cubic directions. Such off-site displacements have been confirmed experimentally; however, being essentially dynamic in nature they cannot account for the static nature of the symmetry-forbidden polarization implied by the macroscopic experiments. Here, in an atomically resolved study by aberration-corrected scanning transmission electron microscopy complemented by Raman spectroscopy, we reveal, directly visualize and quantitatively describe static, 2–4 nm large polar nanoclusters in the nominally non-polar cubic phases of $(Ba,Sr)TiO_3$ and $BaTiO_3$. These results have implications on understanding of the atomic-scale structure of disordered materials, the origin of precursor states in ferroelectrics, and may help answering ambiguities on the dynamic-versus-static nature of nano-sized clusters.

[1] Electronic Ceramics Department, Jozef Stefan Institute, Ljubljana, Slovenia. [2] Interdisciplinary Center for Electron Microscopy, Ecole Polytechnique Fédérale de Lausanne, Lausanne, Switzerland. [3] Group for Ferroelectrics and Functional Oxides, Institute of Materials, Ecole Polytechnique Fédérale de Lausanne, Lausanne, Switzerland. [4] Materials Center Leoben Forschung GmbH, Leoben, Austria. [5] School of Materials and Chemical Technology, Tokyo Institute of Technology, Tokyo, Japan. [6] Department of Materials Chemistry, National Institute of Chemistry, Ljubljana, Slovenia. [7] Present address: Foundation for Research on Information Technologies in Society (IT'IS), Zurich, Switzerland. ✉email: dragan.damjanovic@epfl.ch

Except in the simplest materials, the local symmetry of atomic arrangements is not identical to the nominal crystal symmetry as predicted by phenomenological and first-principles theories or determined experimentally by methods that capture average structure, such as the commonly deployed diffraction techniques[1,2]. Materials that are organized hierarchically respond to external fields with dynamics that depend on the structure and symmetry at all length scales leading to emergent macroscopic behaviours that are nontrivial to interpret[3,4]. One such emergent phenomenon is macroscopic polarization in the nominally non-polar paraelectric phases of ferroelectric materials[5,6]. It has been proposed that the symmetry-forbidden macroscopic polarization results from a collective arrangement of hypothetical polar nanoclusters, which themselves are not allowed by the symmetry of the paraelectric phase[5–14]. While the existence of the polar nanoclusters in the paraelectric phase has been postulated soon after the discovery of ferroelectricity in the first oxide ferroelectric[7], BaTiO$_3$, their presence has not yet been directly confirmed, visualized and characterized on an atomic scale in any classical (non-relaxor) ferroelectric.

A widely considered model of the polar nanoclusters in perovskites assumes correlated displacements of B-site cations along a subset of <1 1 1> cubic directions[9,14]. The model, which does not assume the presence of defects in the material, has been supported by diffuse scattering, diffraction, optical and nuclear magnetic resonance experiments[12,14–18]. However, the dynamic nature of the B-site cation off-site displacement whereby the orientation of polar regions is stable on the time scale of ~$10^{-9}$–$10^{-12}$ s[12,14,19], does not appear to be consistent with the static nature of the macroscopic polarization[5,10,11,20] observed in the paraelectric phases of classical and technologically important ferroelectrics (Ba$_{1-x}$Sr$_x$)TiO$_3$ (BST) and BaTiO$_3$. Excellent dielectric tunability of BST is exploited at microwave frequencies[21,22] while BaTiO$_3$ is the most important capacitor material. An essential parameter in both applications is the dielectric loss. The presence of nanosize polar clusters may significantly contribute to the dielectric loss, especially in the GHz range that is particularly important for telecommunications. A proof of the presence of polar nanoclusters and quantitative details of their atomic structure could significantly alter the interpretation of loss mechanisms in these materials[22–24].

(Ba$_{1-x}$Sr$_x$)TiO$_3$ is a solid solution of the two most studied oxide perovskites. The nominal symmetry of their prototypical structure is cubic $Pm\bar{3}m$. While SrTiO$_3$ is an incipient ferroelectric[25], BaTiO$_3$ is a canonical ferroelectric that transforms on cooling into tetragonal ferroelectric phase at the Curie temperature $T_C = 403$ K, followed by transitions into an orthorhombic and a rhombohedral ferroelectric phase at lower temperatures[26]. For compositions with $x < 0.8$, BST exhibits the same sequence of phase transitions as BaTiO$_3$ and is cubic at room temperature for $x > 0.33$ (see Supplementary Note 1)[27]. For $x > 0.9$ the material possesses relaxor characteristics[28]. A behaviour consistent with a polar or at least a non-centrosymmetric structure has been observed in the paraelectric phases of BST[5,10,29], unmodified BaTiO$_3$[5,9,11,13,20] and in SrTiO$_3$[30,31]. The origin, composition, size and atomic structure of the putative polar nanoclusters in the paraelectric phase of classical perovskite ferroelectrics cannot be inferred in detail from measurements of macroscopic properties. The absence of dielectric dispersion typical for relaxors in BST for $x > 0.1$[28] suggests that the structure of these polar nanoclusters, their dynamics and effect on macroscopic properties is different from those of polar nanoregions in relaxor ferroelectrics[32].

In this study using aberration-corrected scanning transmission electron microscopy (STEM) in combination with energy-dispersive x-ray spectroscopy (EDXS) and supported by Raman spectroscopy, we give a direct atomistic evidence for polar nanoclusters in paraelectric BST and BaTiO$_3$, and discuss quantitatively clusters' atomic-scale structure and polar properties. In addition to answering this long-standing question, the results of the study will have an impact on understanding the dielectric tunability and loss mechanisms in the paraelectric phases of ferroelectrics[22–24].

## Results

**Evidence for non-cubic nanoclusters and their polar character.** We start by presenting the evidence for nanoclusters with a symmetry lower than the expected cubic in (Ba$_{0.6}$Sr$_{0.4}$)TiO$_3$ (BST6040). STEM measurements (see "Methods") were made at an ambient temperature where BST6040 is nominally cubic ($T_C \approx$ 273 K on cooling, see Supplementary Figure 1). Figure 1a shows a high-angle annular dark-field (HAADF) Z-contrast image of a BST6040 grain viewed along the [0 0 1]$_{pc}$ zone axis (all orientations are indexed as pseudo cubic—pc). Arrows in Fig. 1a illustrate the direction and magnitude of displacement of Ti atoms (B sites) from their expected ideal cubic positions set by the projected centre of the A-site sublattice (for details see Supplementary Note 2). The longest arrows correspond to a displacement of ~15 pm. Groups of atoms with correlated orientation of displacements are highlighted by arrows' colour in Fig. 1b. These groups of atoms form ~2 to 4 nm large clusters. Importantly, our data suggest that a given nanosize cluster is not isolated from neighbouring clusters by the material's cubic matrix (see caption of Fig. 1a, b); the whole structure rather resembles a "slush-like structure"[33,34], so that clusters interact directly with each other. Note that those regions with small atom displacements in the plane of the image might exhibit larger displacements in the direction perpendicular to the image plane, and thus probably also do not possess the cubic structure. The positions of oxygen atoms are visible in the annular bright-field (ABF) images in Fig. 1c, d. As illustrated in Fig. 1e, oxygen atoms, too, are displaced (Fig. 1c) from ideal positions (Fig. 1d) expected in the cubic perovskite cell. The local structure of BST6040 is, therefore, nowhere over the examined area ideally cubic, even though the average structure of the material, as determined by powder x-ray diffraction at room temperature, is cubic perovskite (see Supplementary Figure 1a).

Further evidence for a local symmetry-breaking and presence of non-cubic nanosize clusters in BST6040 is obtained from the analysis of Raman spectra, as shown in Fig. 1f (see also "Methods"). The evolution of the Raman spectra suggests three structural regimes in BST6040: (i) a ferroelectric phase from low temperatures up to 288 K, (ii) a cubic paraelectric phase with polar static disorder and ferroelectric-like distortions up to 573 K and (iii) a cubic paraelectric phase with dynamic disorder above 573 K. Below 268 K, the first-order Raman modes dictated by the Raman selection rules for the ferroelectric phase of BST6040 (marked by arrows in Fig. 1f) dominate[29]. These modes appear when the long-range ferroelectric order is established in a system at least in one crystallographic direction[35]. From 268 to 288 K, the intensity of the ferroelectric modes decreases, and two broad modes at ~220 and ~550 cm$^{-1}$ (marked by asterisks and dotted lines) become evident. These two modes are ascribed to additional Raman scattering activated by (static) broken translational symmetry caused on the short range by lattice defects (i.e. Sr$^{2+}$ substitution). Such static, disorder-activated modes appear in the vicinity of the first-order modes associated with the vibrations of lighter atoms (i.e. BO$_6$ octahedra)[36–38]. These modes are different from second-order Raman modes visible in the case of dynamic disorder: their scattering mechanism is essentially first

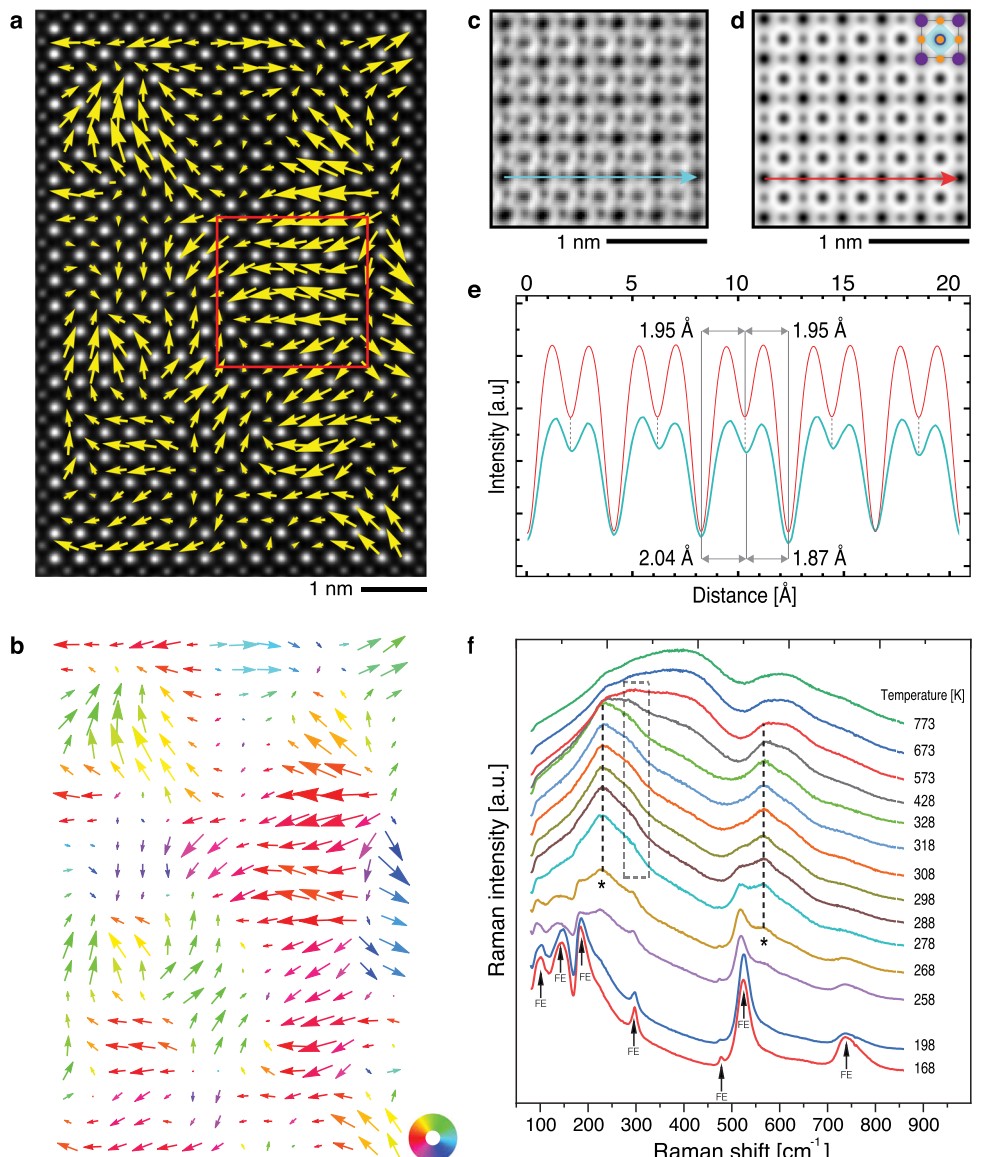

**Fig. 1 Evidence of polar nanoclusters in cubic BST6040. a** HAADF image of a 14 × 21 unit-cell area viewed along the [0 0 1]$_{pc}$ zone axis. The bright spots correspond to the (Ba,Sr) columns (A site) and smaller, paler spots to the Ti columns (B site). Arrows show displacement of B-site atoms measured as the deviation of the experimental position of B sites from the centre of the four neighbouring experimental A-site positions (see Supplementary Note 2). The length and orientation of arrows indicate magnitude and angle of the relative displacement of Ti from the ideal position. **b** The same as in **a** with the colour of arrows indicating regions of coherent displacements, highlighting different nanoclusters. **c** Experimental ABF image from the region marked by the red rectangle in **a**. **d** Simulated ABF image of cubic BST6040. **e** Intensity profiles along the arrows shown in **c**, **d**: cyan, experimental ABF image; red, simulated ABF image with cubic phase. Displacement of oxygen atoms from the ideal cubic sites is visible in the cyan profile. **f** Raman spectra of BST6040 as a function of temperature. The modes marked by arrows are characteristic of the ferroelectric (FE) phase and those marked by asterisks indicate broken translational symmetry caused, on the short range, by lattice defects (i.e. $Sr^{2+}$ substitution). The first-order phonon mode at 300 cm$^{-1}$ (enclosed by the dashed square) indicates that the ferroelectric-like distortion is present on the short-range scale well within the paraelectric phase (above $T_C$ = 273 K). The presence of polar and disorder-activated modes up to 573 K indicates that transition to purely dynamic disorder is attained only above this temperature (see text for details).

order. In the presence of static disorder, such modes coexist with the first-order (zone-centre) phonons and tend to be more evident upon weakening of the latter, as seen in Fig. 1f at >268 K. The disappearance of the long-range ferroelectric phase and the transition to the regime (ii) of static, defect-induced disorder is clearly seen at 288 K by the disappearance of the ferroelectric modes at ~200 and ~515 cm$^{-1}$, which is in good correspondence with the Curie temperature of BST6040. From this temperature onwards, the two disorder-activated modes coexist with the first-order mode at 300 cm$^{-1}$ (dashed square in Fig. 1f), which persists

in the Raman spectrum up to 573 K. This mode is related to the relative displacement of the A- and B-site sublattices with respect to the oxygen sublattice in ABO$_3$ perovskites[37] and, as it indicates asymmetry within the BO$_6$ octahedra[39], it is active only in the polar phase. In other words, the mode suggests the presence of a non-zero dipole moment in the lattice up to 573 K (roughly 300 K above the $T_C$). First-order scattering >288 K, signalled by the presence of the 300 cm$^{-1}$ mode, is the result of the coupling of normal modes with the defect-induced lattice distortion, leading to a scattering contribution from a large part of the Brillouin

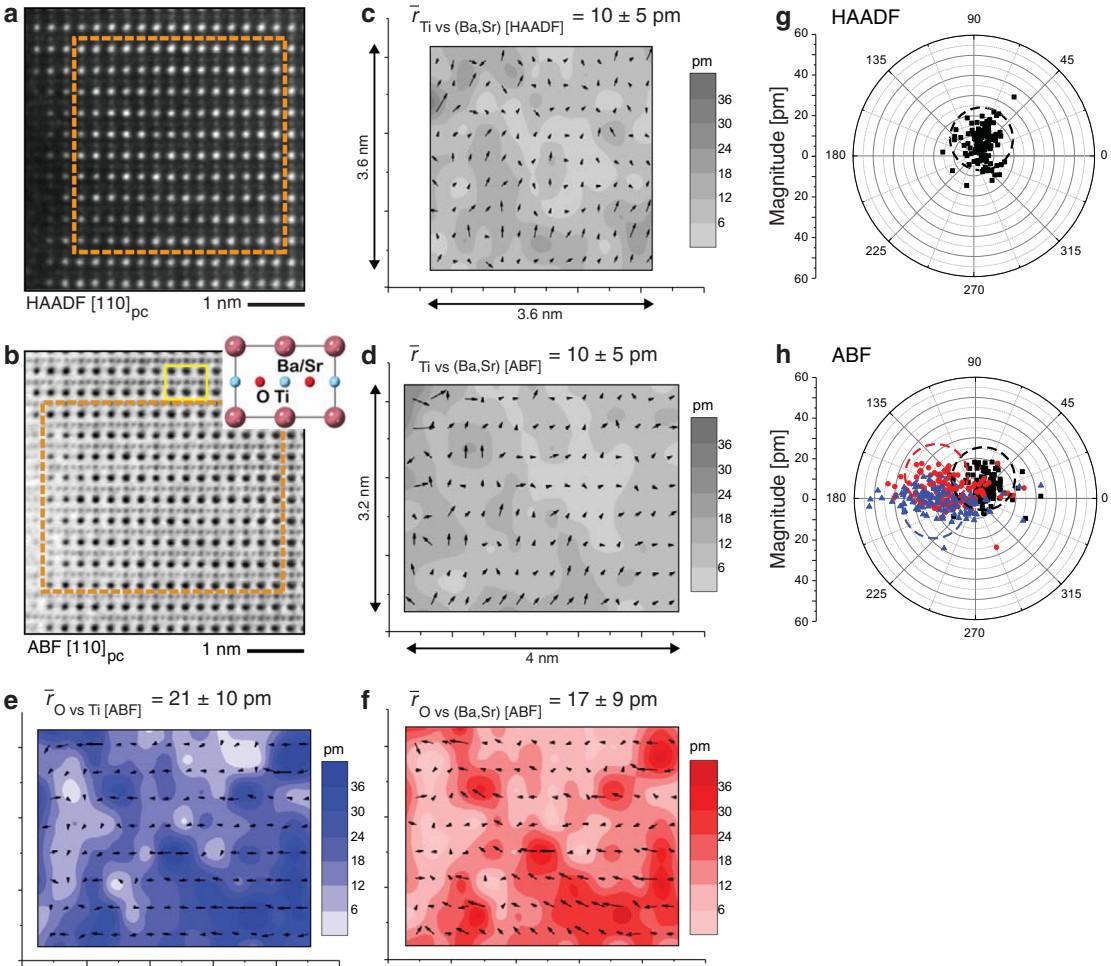

**Fig. 2 Analysis of atomic displacements in BST6040. a** HAADF image and **b** ABF image, viewed along the $[110]_{pc}$ zone axis of an ~15-nm-thick BST6040 grain. Dashed rectangles mark areas where displacements were determined. The inset in **b** shows the perovskite unit-cell viewed along $[110]_{pc}$ axis, one of which is marked by the yellow rectangle. **c–f** Displacements of Ti vs (Ba, Sr), O vs Ti and O vs (Ba, Sr) determined from HAADF (**c**) and ABF (**d–f**) images with respect to positions of these atoms in an ideal cubic phase with the corresponding average displacements ($\bar{r}$). Arrows represent the direction and magnitude of the displacements and colours represent the magnitude of displacement given as contour plots (see the colour scale on the right of each image). **g**, **h** Compilation of the data presented in **c–f** in the form of polar plots, which delineate the orientation of atom displacements. Each symbol and its position in the polar plot correspond to one arrow, with black squares representing Ti vs (Ba, Sr), blue triangles O vs Ti and red circles O vs (Ba, Sr) displacements. The polar plots show that the symmetry of the local atomic structure is not cubic, is most likely polar and possibly possesses monoclinic or lower symmetry. Dashed circles mark areas where the majority of the individual type of displacements are present.

zone, within a length scale dictated by the static disorder[40]. Such effects were already seen in $KTaO_3$-incipient ferroelectric[40] and even in strained $BaTiO_3$ thin films[41] and were attributed to the presence of static disorder-driven ferroelectric regions in nominally paraelectric phases, and stress-induced ferroelectric phase stabilization, respectively. The static disorder is also on the basis of the broad appearance of the $300\,cm^{-1}$ mode, which indicates coherence of the ferroelectric phase on the short range (i.e. a few nanometres) rather than on the long range. Above 573 K, the transition from static disorder to dynamic disorder is evident from the complete disappearance of modes related to the polar state together with the disorder-activated modes, and the concurrent onset of a broader spectral signature. The broad spectrum $>573\,K$, in fact, is due to second-order Raman scattering reflecting the phonon density of states of BST6040. Such spectral signature is due to the time-average of symmetry-breaking events with random wavevector[35], such as dynamic uncorrelated displacements of the Ti cation in random directions. The persistence of first-order phonon modes at $300\,cm^{-1}$ above $T_C$, which are only allowed in the polar state along with the

disorder-activated modes centred at 220 and $550\,cm^{-1}$ is a clear evidence that a defect-driven ferroelectric-like distortion is present in the bulk paraelectric phase of BST6040. Hence, the Raman spectra not only confirm the presence of nanoclusters in the paraelectric phase but, importantly, also indicate that these clusters are polar.

**Quantification of relative atomic displacements.** We next take a closer and quantitative look into the structure of nanoclusters in BST6040 on the atomic scale. To quantify the structural distortion, the relative displacements of atoms [Ti vs (Ba,Sr), O vs (Ba, Sr) and O vs Ti] were determined with respect to expected atomic positions in the cubic phase (Fig. 2). The displacements were calculated from positions of individual atom columns extracted from HAADF and ABF images using 2D Gaussian fit as described in ref. [42] (see also "Methods" and Supplementary Note 2). For this calculation, we used HAADF and ABF images taken along [1 1 $0]_{pc}$ zone axis, as shown in Fig. 2a, b. The effects of sample mistilt and sample thickness on the displacement measurements were

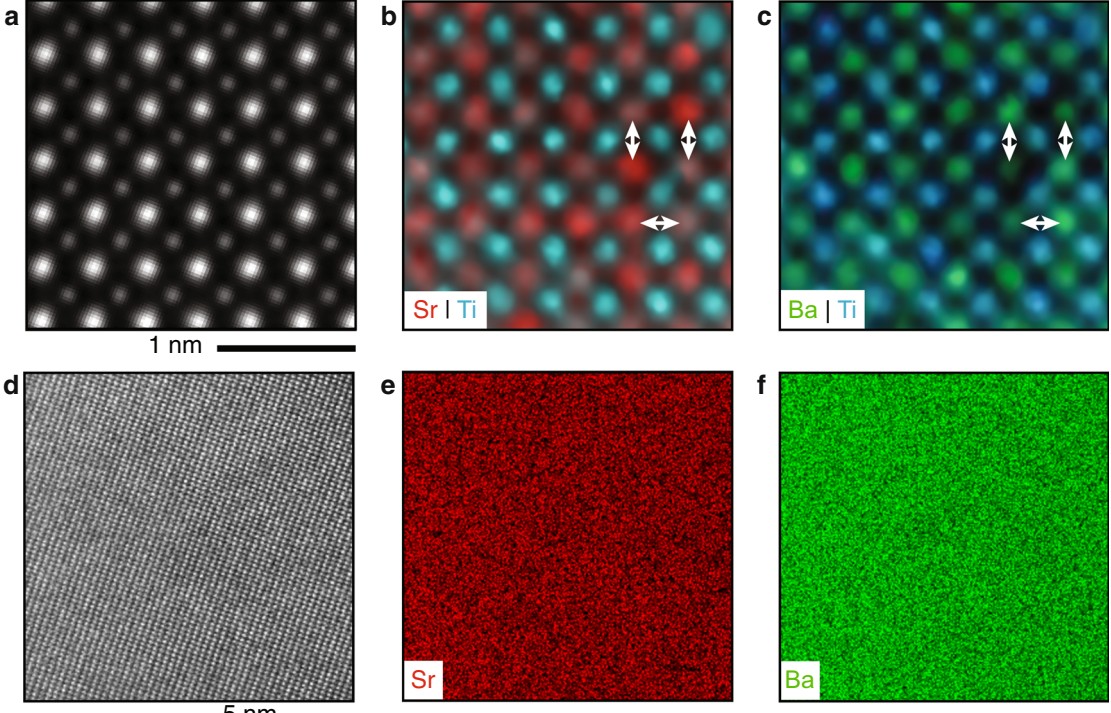

**Fig. 3 Nanoscale and atomically resolved chemical analysis of BST6040. a** HAADF-STEM image viewed along the $[0\,0\,1]_{pc}$ zone axis of an ~2 nm region from Fig. 1a. The corresponding EDXS intensity maps of **b** Sr/Ti and **c** Ba/Ti. The arrows indicate Sr (**b**) and Ba (**c**) rich (high intensity) and poor (low intensity) columns. **d** HAADF-STEM image of a sample segment about ~20 × 20 nm² in lateral dimensions and corresponding EDXS intensity maps of **e** Sr-L and **f** Ba-L lines. The images illustrate chemical homogeneity on several nanometre scale and heterogeneity on unit-cell and column-to-column scale, while the Ba/Sr 60/40 stoichiometry is preserved on all levels (see Supplementary Note 4).

considered and taken into account as explained in Supplementary Note 2. A view along $[1\,1\,0]_{pc}$ zone axis is convenient for the quantitative analysis of displacements as it gives access to columns consisting only of Ti atoms and columns containing only those O atoms that lay in the plane of the $O_6$ octahedra, thus providing a more reliable determination of relative displacements of ions with a low atomic number than the view along the $[0\,0\,1]_{pc}$ zone axis.

The displacements determined from experimental HAADF and ABF images of BST6040 are presented in Fig. 2c and d–f, respectively. The examined area (~4 × 4 nm²) captures roughly one nanocluster. The same information is compiled in the accompanying polar plots (Fig. 2g, h) where the direction of displacements is clearly delineated. The average measured displacements $\bar{r}$ are in the range from ~10 pm (for Ti vs (Ba, Sr)) to ~21 pm (for O vs Ti). There are two key implications of these results. First, Ti vs (Ba, Sr) displacements measured from HAADF and ABF images are non-zero with similar direction and magnitude indicating non-cubic symmetry. In addition, the different magnitudes and directions of $O^{-2}$ anions displacements with respect to those of $(Ba, Sr)^{+2}$ and $Ti^{+4}$ cations measured from ABF image indicate separation of gravity centres for positive and negative charges, i.e., suggesting a non-zero dipole moment in the plane of the image within the examined area. The 2D displacements qualitatively resemble those in simulated orthorhombic $BaTiO_3$ where, unlike in rhombohedral and tetragonal symmetries, O vs Ba displacements are concentrated away from the origin of the coordinate system (see Supplementary Notes 3 and 7), possibly indicating monoclinic or lower symmetry of the local 3D structure in BST6040. We note that the magnitude of displacements is comparable to those calculated by reverse Monte Carlo refinements from multiple measurement techniques (see Supplementary Note 2)[43].

**Chemical analysis**. To see whether the polar nanosize clusters in BST6040 originate from the segregation of ferroelectric, $BaTiO_3$-rich regions with a $T_C$ above room temperature, we carried out atomic-resolution chemical analysis with EDXS (Fig. 3). The chemical heterogeneity is visible, as expected, only on the level of individual atom columns (Fig. 3a–c), but there is no substantial chemical heterogeneity over the larger examined areas of ~20 × 20 nm² (Fig. 3d–f). For a quantitative estimation of the local stoichiometry of the sample, we modelled cubic BST6040 with a random distribution of Ba and Sr (for details see Supplementary Note 4) and compared the intensities of atomic columns in the simulated images with those in experimental images. This analysis shows that there is no significant departure of Ba/Sr ratio in our samples from the random distribution of Ba and Sr atoms with 60/40 stoichiometry.

That the material is chemically homogeneous on tens of nanometre scale and the chemical heterogeneity is present only on column-to-column and unit cell-to-unit cell level, as expected for a random Ba/Sr distribution with the overall 60/40 stoichiometry, was further supported by the observation of macroscopic polarity in samples prepared by different procedures (see Supplementary Note 1). It can be, therefore, stated that the polar nanoclusters are the property of BST6040 and not a question of the degree of mixing and interdiffusion of Ba and Sr in a given sample, which could be processing dependent.

**Constraints of the 2D analysis**. To explore the constraints of the 2D analysis of STEM data on the determination of the polar nanoclusters' size and relative displacements of atoms, which take place in 3D, we modelled the displacements in a volume consisting of a polar structure embedded within a cubic matrix (details in Supplementary Note 5 and Supplementary Figure 11).

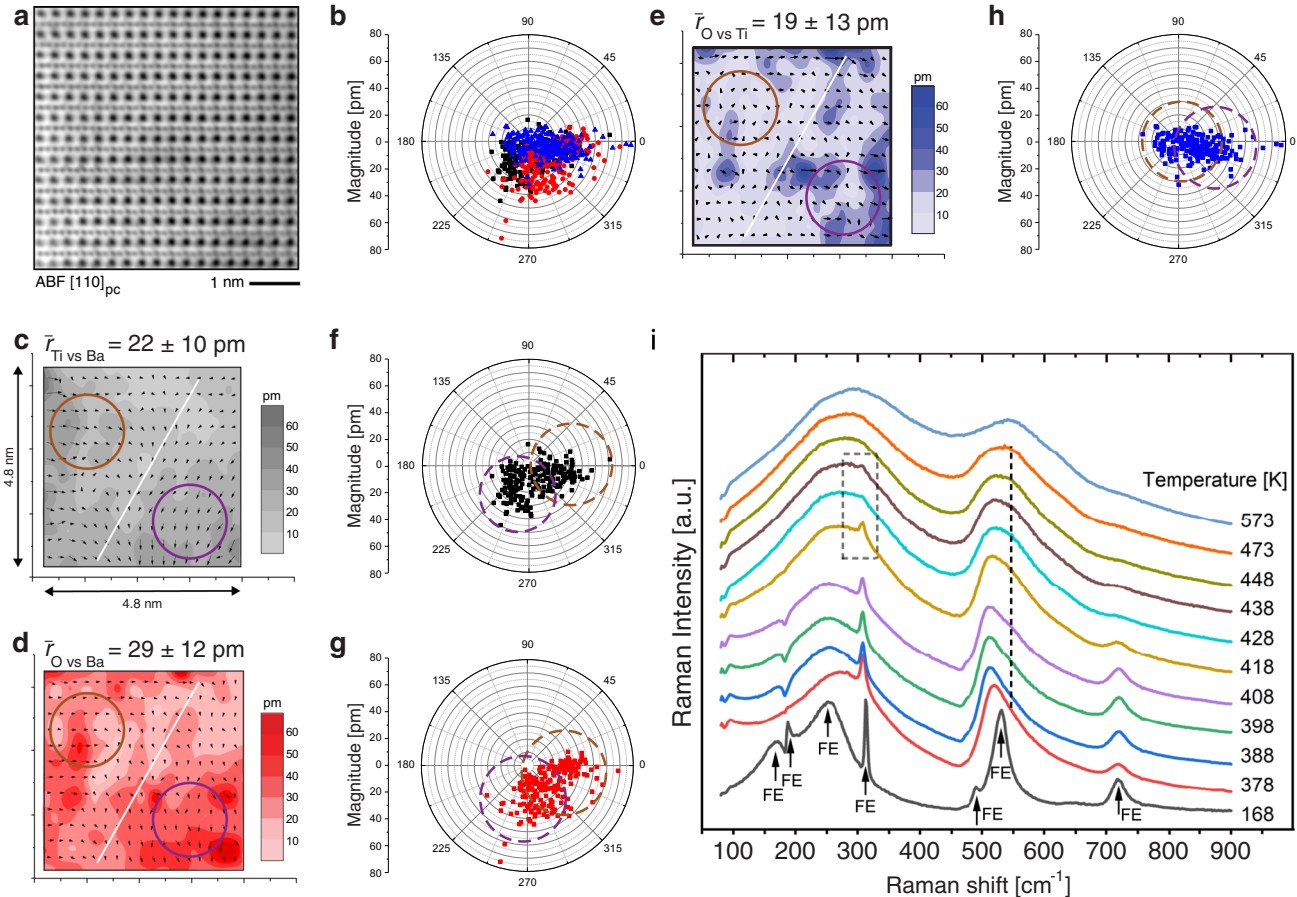

**Fig. 4 Atomic displacements and Raman analysis of BaTiO₃ at 473 K. a** ABF image of cubic phase of BaTiO₃ viewed along the [110]$_{pc}$ zone axis that was analysed to determine the displacements with **b** composite polar plot, which compiles data presented in **c**–**h**. The black squares represent Ti vs Ba, blue triangles O vs Ti and red circles O vs Ba displacements. **c** Ti vs Ba, **d** O vs Ba and **e** O vs Ti relative atomic displacements in a BaTiO₃ grain. Arrows represent the direction and magnitude of the displacements (see the colour scale on the right of each image). The interface between the two polar nanoclusters is indicated by the diagonal lines. The brown and violet full circles indicate representative areas of each cluster. **f**–**h** The same data as in **c**–**e**, respectively, presented in polar plots, delineating the orientation of the atomic displacements. Each symbol and its position in the polar plot correspond to one arrow. Dashed brown and violet circles are guides to eye to indicate two polar nanoclusters. **i** Raman spectra of BaTiO₃ as a function of temperature. First-order phonon modes associated with the ferroelectric phase (FE) are indicated by arrows and (e.g. the mode at 300 cm⁻¹, marked by dashed square) are identified up to 438 K (i.e. above $T_C$ of ≈403 K). From 438 to 473 K, the modes belonging to the static disorder are still present (marked by dotted line). Above 473 K, only second-order modes resembling the phonon density of states of BaTiO₃ are recorded. Hence, the static (polar) disorder is present not only in BST6040 but also in BaTiO₃, up to at least 438 K and a transition to the dynamic (non-polar) disorder occurs >473 K.

We conclude that the polar displacements in an embedded volume with a thickness up to few nanometres can be reliably detected by the procedures used in this work. It was found, in fact, that the atomic displacements associated with deeper regions in the sample (>5−10 nm) have a marginal influence on the displacement orientation viewed in the STEM images (see Supplementary Figure 12 in Supplementary Note 5). Furthermore, since the nanosize clusters appear to be randomly distributed, are roughly isotropic in size (Fig. 1) and we do not see on the surface individual nanosize polar clusters with coherent polarization >∼4 nm, we can state that the nanosize clusters observed in BST6040 are roughly 2–4 nm large in three dimensions.

**Polar nanoclusters in cubic BaTiO₃.** We next present and discuss the evidence for polar nanoclusters in the unmodified BaTiO₃. This part of the investigation was further complicated by the need to heat samples in situ above the $T_C$ (≈403 K) to reach

the cubic phase (see "Methods"). At 473 K we observe a region composed of two polar nanoclusters (Fig. 4). Two populations can be identified for each of the three pairs of relative atomic displacements (Ti vs Ba, O vs Ba and O vs Ti), further confirming that our method in determining quantitatively displacements is sufficiently robust to identify and resolve neighbouring nanoclusters with different orientation of displacements and associated polarization. The interface between the two populations, spreading roughly through the middle of the examined region, defines the two nanoclusters. The O vs Ti displacements for the two populations are colinear but with different magnitudes. One population of O vs Ti displacements is centred around zero, indicating a low magnitude of displacements' projection in the plane of the image for that nanocluster. The Ti vs Ba and O vs Ba displacements are making roughly an angle of 90° for both populations and are non-zero. To inspect whether these displacements could correspond to the tetragonal, orthorhombic or rhombohedral polar structure of BaTiO₃, we have modelled these

structures and compared the associated displacements (Supplementary Note 3 and Supplementary Figure 7) with the experimental data (Fig. 4). The symmetry of the experimental displacements in the nominally cubic phase are qualitatively different than in any of the polar phases of $BaTiO_3$. The experimental data are thus consistent with the presence of two polar clusters in the examined region, each possessing an overall monoclinic (or lower) symmetry. The low symmetry of the nanoclusters and their size of roughly 2–4 nm are similar to those found in BST6040. The clusters exhibit coherent relative displacements of cations and oxygen, suggesting their overall polar character. We note that measured displacements agree reasonably well with those predicted by the Density Functional Theory computations that investigated the effects of the addition of various defects to a cubic $BaTiO_3$ supercell (see Supplementary Note 3).

At 573 K, all displacements are smaller than at 473 K, but are still non-zero, with the scattering smaller than at 473 K, and the average structure resembling cubic, although some tendency toward non-cubic distortion can be discerned in Ti vs Ba relative displacements (see Supplementary Figure 9 in Supplementary Note 3). These data are in agreement with the results obtained by acoustic emission study[44], which indicated a change in the structure of hypothetical polar nanoclusters in the temperature range between 500 and 550 K.

The Raman spectra of a $BaTiO_3$ sample (Fig. 4i) confirm the conclusions of the STEM atomic-scale data analysis and suggest a behaviour similar to that in BST6040. First-order Raman modes (marked with arrows) associated with the ferroelectric order (e.g. the mode at 300 cm$^{-1}$) are observed at least up to 438 K (i.e. ~40 K above $T_C$), whereas disorder-activated modes (seen here clearly at ~520 cm$^{-1}$—the disorder-related mode at ~270 cm$^{-1}$ is convoluted with strong first-order modes up to $T_C$) are detected up to 473 K. This temperature thus marks the approximate upper limit of static disorder in $BaTiO_3$, suggesting polar nanoclusters at least up to 438 K. The disorder in unmodified $BaTiO_3$ is probably activated by lattice defects such as oxygen or metal (Ba and Ti cations) vacancies[36,38]. Above 473 K, transition to dynamic disorder is signalled by the fully second-order Raman signature. Considering the expected differences in the temperature measurements between in situ STEM and Raman measurements, the transition temperatures between the static and the dynamic disorder obtained by the two techniques agree rather well.

**Freezing of polarization orientation**. The model by Comes et al.[9] of disordered cubic phase in perovskite $BaTiO_3$ and $KNbO_3$, which is based on diffuse X-ray and electron scattering, suggested off-site dynamic displacements of B-site cations along <1 1 1> directions. It was proposed that these displacements are correlated on a short-range forming chains and anti-chains along <0 0 1> axes. A random distribution of those chains gives on average non-polar, cubic symmetry. The length of the polar chains was predicted to be between 4 and 10 nm[9,45]. Subsequent studies reached similar conclusions, pointing out short duration (nanoseconds to picoseconds) of correlated displacements along a given direction[12,14–16]. Off-site displacements have been confirmed experimentally in BST by refinement of atomic positions obtained from neutron scattering, x-ray absorption and electron diffraction data[43]. However, a recent study of $BaTiO_3$ by synchrotron X-ray diffraction indicated that the diffuse scattering can be explained by anharmonic phonons and stated that polar chains such as those proposed by Comes et al. cannot be stabilized for more than a few picoseconds because of a too low energy barrier among different off-site positions[19].

Our results add thus an additional layer of complexity to this long-standing problem: we show that the atomic structure of the nanosize polar clusters in cubic BST and $BaTiO_3$ is essentially static, at least on the scale of hundreds of seconds (time needed to make STEM or carry out macroscopic property measurements[5]). The question can then be posed what stabilizes the orientation of polarization within polar nanoclusters? We propose that the driving force for stabilization of local polarity direction in BST and $BaTiO_3$ may be provided by local strains within and around clusters; we have indeed experimentally observed evidence of such strains (see Supplementary Note 6 and Supplementary Figures 13 and 14). In BST the strains may be due to the size difference of Sr and Ba and could be related to correlated atomic displacements from ideal cubic positions. These unit-cell and cluster-level strains may freeze B cation within a nanocluster in one of the eight split sites or at least lead to unequal times the cation spends in different off-site positions[46]. Furthermore, it is known that the size difference of Ba and Sr leads to unequal Sr–O and Ba–O distances in BST, while the off-site displacement of Ti depends on the number of Sr cations around it[43]. A similar role could be played in $BaTiO_3$ by impurities, such as O, Ba and Ti vacancies, and possibly reduced $Ti^{+3}$ cations on the B perovskite site (note that $Ti^{+3}$ and host $Ti^{+4}$ possess different size[47]). Significantly, the presence of strains disrupting centrosymmetricity has also been clearly demonstrated by the activation of first-order Raman modes in the nominally paraelectric phase of both BST6040 and $BaTiO_3$.

While for BST it appears clear that the strain can be locally provided by the large concentration of Sr, the situation in $BaTiO_3$ is less obvious. It is important to note that a large concentration of Sr is not essential for the stabilization of polarization within nanoclusters even in BST. We have evidence from macroscopic measurements of the formation of polar nanoclusters in the paraelectric phase of $Ba_{0.975}Sr_{0.025}TiO_3$ single crystals and ceramics[48], which have 16 times smaller concentration of Sr than the BST6040 investigated here.

We examine next the point defect concentration of $BaTiO_3$ in detail. For the synthesis of our samples, we have used standard electronic grade powder of $BaTiO_3$ with 99.95% (metal basis) purity (see Supplementary Note 1). Based on the information presented in Supplementary Note 1, we estimate one metal defect separated in three dimensions from a neighbouring defect by 12–13 unit cells or about 5 nm. This distance agrees very well with the length of regions with coherent polarization in the paraelectric phase of $BaTiO_3$ proposed by Lambert and Comes[45]. In other words, since the polarization within nanoclusters is a correlated effect, freezing of ionic displacements in one unit cell due to the presence of a defect may have an effect on the dipole direction in the neighbouring cells, freezing the polarization orientation in the whole nanocluster. Other evidence in the literature suggests that this hypothesis is plausible. For example, it has been shown that the forbidden second harmonic signal in paraelectric $BaTiO_3$ is stronger in ceramics than single crystals, and this has been correlated to a larger concentration of defects in the former[13]. Darlington and Cernik showed a correlation between a tetragonal distortion in paraelectric $BaTiO_3$ with a small concentration of point defects[49]. Wada et al. visualized the presence of a disorder-activated Raman mode in the paraelectric phase of unmodified $BaTiO_3$ single crystals, and ascribed it to lattice defects[50]. We have observed evidence of this disorder-activated Raman mode also in this work (Fig. 4).

The stability of $BaTiO_3$ under the electron beam was also investigated. Specifically, we tested whether a significant number of Ba vacancies could be produced by prolonged irradiation of the samples with electrons. As shown in Supplementary Note 7, we

find that under experimental conditions used in this study, defects are not produced in $BaTiO_3$ during HAADF experiments.

Finally, we investigated whether the electron beam influences atomic displacement measurements. From the results presented in Supplementary Note 7, we can conclude that polar nanoclusters are not formed or modified during observation of $BaTiO_3$ sample in STEM experiments.

## Discussion

The study provides direct atomic-scale evidence and structural details of nanoscale objects whose hypothetical presence in the paraelectric phase of perovskite ferroelectrics has been discussed for decades. Static, roughly 2–4 nm large polar nanoclusters are revealed above $T_C$ in BST6040 (at room temperature) and $BaTiO_3$ (at 473 K). In BST6040, these regions are not associated with Ba segregation.

Our results suggest that the direction of polarization in polar nanocluster is not along polar directions of tetragonal, orthorhombic or rhombohedral phases, but rather along an arbitrary direction, with the individual unit cells displaying at most a monoclinic symmetry. The polar nanoclusters cover most of the investigated areas. Therefore, only the average structure of the material is cubic, but in contrast to previous thinking, our study shows that the direction of polarization within a given polar nanocluster is stable on the time scale of measurements (tens to hundreds of seconds). The probable reason for the stabilization is the presence of local strains, which originate from the size difference between additives, impurities, vacancies and host ions. These polar nanoclusters are consistent with objects that can explain symmetry-forbidden details in the Raman spectra of the paraelectric phase of barium-titanate-based perovskite ferroelectrics. Measurements in $BaTiO_3$ show that clusters exhibit dynamic nature at sufficiently high temperatures (573 K).

The revealed polar nanoclusters are active under an external electric field and contribute to the polarization of the material[5,51]. The identification of polar nanoclusters, their size, spatial distribution and properties, which are presented in this work could have an impact on modelling dielectric tunability and high-frequency losses in this family of materials which are widely used in capacitors and microwave communications.

## Methods

**Material preparation**. $(Ba_{1-x}Sr_x)TiO_3$ ceramics ($x = 0$–1, with steps of 0.1) were synthesized by solid-state reaction of $BaTiO_3$ and $SrTiO_3$ precursors. All investigations for this study were made on composition with $x = 0.4$ (BST6040) and $x = 0$ ($BaTiO_3$). BST6040 exhibits $T_C$ at 273 K during cooling; therefore, the material's nominal structure at room temperature is cubic, centrosymmetric $Pm3m$. The grain size of ceramics is >5 μm. Thus, every sample examined by high-resolution STEM is a single crystal. For more details see ref. [5] and Supplementary Notes 1 and 5.

**Electron microscopy**. The BST6040 samples for STEM were prepared by a combination of mechanical polishing, followed by argon ion beam milling to electron transparency. $BaTiO_3$ samples were analysed in the powder form, which was ground, annealed at 900 °C for 2 h in air and slowly cooled down to room temperature to eliminate residual strains. STEM imaging and EDXS were performed on a double Cs-corrected Thermo Scientific Titan Themis 60–300 (equipped with Super-X EDX system) and a probe Cs-corrected Jeol ARM 200 CF (equipped with Centurio EDX system). The thickness of the STEM samples was estimated from low-loss EELS spectra collected by a Gatan Quantum ER Dual EELS spectrometer.

HAADF and ABF images were simultaneously acquired in the form of image series that underwent an image alignment procedure to produce an averaged image with reduced statistical image noise (see Supplementary Note 2 for more details).

The central position associated with each atomic column was localized on the averaged HAADF and ABF images using a two-dimensional Gaussian fitting procedure, and displacements for the B- and O-site columns were determined by measuring their displacements from ideal cubic positions. For experimental details of imaging and atom spacing measurement, see Supplementary Note 2.

In situ heating experiments of $BaTiO_3$ were performed using a heating TEM holder made by Protochips (Fusion model). Images were taken at room temperature, 473 K and 573 K.

STEM image simulations were performed using quantitative image simulation code (QSTEM)[52] with multislice method and frozen phonon approximation. Details of calculations are described in Supplementary Notes 2 and 5 for BST and in Supplementary Note 3 for $BaTiO_3$.

**Raman spectroscopy**. Raman measurements were carried out in a LabRAM 300 spectrometer (Horiba Jobin Yvon, Villeneuve d'Ascq, France) using an Nd: YAG solid-state laser with a wavelength of 532 nm in a backscattering geometry. The laser light was focused on the sample surface by means of a long working distance ×100 objective (with NA 0.8, LMPlan FI, Olympus, Tokyo, Japan). The effective power at the sample surface was kept <3 mW. The spectra were collected with a Peltier-cooled charge-coupled device and visualized in the commercial software environment (Origin 2018b, OriginLab Corp., Northampton MA, USA) after correcting for the Bose–Einstein population factor. Temperature-dependent Raman measurements were carried out in a Linkam (THMS600, Linkam, Tadworth, UK) temperature-controlled stage.

## Data availability

The data that support the findings of this study are available from the corresponding author upon reasonable request.

Published online:

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

## Acknowledgements

A.B, T.R. and G.D. acknowledge funding from the Slovenian Research Agency within programmes P2-0105 and P2-0393 and project J2-2497. Brigita Kmet and Daniele Laub are acknowledged for the STEM sample preparation. E.O. thanks Duncan T.L. Alexander and Sorin Lazar for constructive discussions. S.H. and D.D. acknowledge financial support from the Swiss National Science Foundation (No. 200021-159603) and Alberto Biancoli for discussions and help in samples preparation. V.K.V. and M.D. acknowledge support from the Austrian Science Fund (FWF) Project P29563-N36. M.D. acknowledges funding from the European Research Council (ERC) under the European Union's Horizon 2020 research and innovation programme (grant agreement no. 817190). Ronald J. Bakker (Montanuniversität Leoben, Chair of Resource Mineralogy) is acknowledged for providing access to the Raman equipment.

## Author contributions

S.H. prepared samples, measured and interpreted macroscopic properties. T.H. prepared samples using different techniques and investigated the effects of defects and micro-structure on the forbidden polar properties. V.K.V. and M.D. performed Raman measurements and analysis. T.R. critically commented on the results, the data analysis and the text. E.O., A.B. and G.D. performed all atomic-resolution investigations and analysed and interpreted STEM data. A.B., G.D., E.O., V.K.V., M.D. and D.D. wrote the manuscript with input from all coauthors. D.D. initiated the study and developed the concept.

## Competing interests

The authors declare no competing interests.
