## [Peer Review File · Nature Communications]

REVIEWER COMMENTS

Reviewer #1 (Remarks to the Author):

This work seems to be done well with a lot of attentions to reliability and accuracy of experimental results. Existence of nano-sized polar region in ferroelectric perovskites is one of the field of debate in current metal oxide functionality. From this viewpoint, this paper seems to be on-time. However, the reviewer wonder if technical importance of this paper is very high to justify publication. This paper focuses on atomic scale disorder. As authors mentions, the averaged chemical composition and properties of $(\text{Ba,Sr})\text{TiO}_3$ solid solution is very homogeneous but locally disorders because of random distribution of Ba and Sr. Indeed, the authors obtained atomic displacement image. However, audiences may not be very surprised with their results because everybody knows distribution of Sr and Ba in $(\text{Ba,Sr})\text{TiO}_3$ should be random and unit cell dimension with Sr and Ba must not be same. There may be some polarization because of the random distribution of Sr and Ba. However, if we look at usage or application of $(\text{Ba,Sr})\text{TiO}_3$, its averaged bulk properties appears. Hence, it seems that this work is just for visualization of what we can imagine. Regarding disorder in BaTiO_3 , authors ascribed the local inhomogeneity to defects. The conclusion sounds too straightforward. Reviewer would like to point a serious difficulty relating to this report. That is, the researchers cannot know the defect concentration in the sample under STEM/TEM observation. As STEM utilizes electron beam with high brightness, there is a chance to induce defects by irradiation during observation. It is, indeed, difficult to quantify defect concentration induced by TEM observation. It is also difficult to know the defect concentration in the original sample before TEM study. If the authors' sample possesses highly insulating properties, defect concentration in their sample was low enough and likely on the order of 10^{16} cm^{-3} or further less. This number corresponds to approximately one defect among one million atoms. That means defect concentration in BaTiO_3 must be negligibly small compared to Sr fraction at A-site in $(\text{Ba,Sr})\text{TiO}_3$. With this consideration, attributing 'observed' inhomogeneity to defect seems unrealistic. On the other hand, authors' conclusion may be acceptable if defect concentration in BaTiO_3 is a few orders of magnitude higher than reviewer's guess. If this is the case, BaTiO_3 under TEM observation is unlikely similar to ordinary bulk BaTiO_3 .

One possible response from the authors should be adding consideration of the defect concentration. The reviewer would like to ask the authors, how much defect concentration has to be assumed to enable them to detect defect related structural relaxation with their techniques. As atomic displacement around point defects have been studied by density functional theory calculations, I suppose the authors can estimate possible STEM images with assumed defect concentration. If they cannot explain observed displacements in BaTiO_3 with reasonable number of defects, that may be an indication that defects are formed during sampling or observation. If the actual defect concentration in BaTiO_3 under STEM observation was much higher than that in ordinary bulk BaTiO_3 , conclusion of this paper may be a cause of misleading for defect structure.

I would be happy to review the revised version if further quantitative discussion is given for defect structure and defect concentration in BaTiO_3 in the revised version.

Reviewer #2 (Remarks to the Author):

The authors report the experimental observations of polar domains in Sr-doped BaTiO_3 systems in the (macroscopically) cubic phase. The properties and structures of these system have been a subject of intense exploration since the early fifties, yet only recently have the advances in aberration corrected

STEM allowed to visualize these structures directly. The authors perform a careful study exploring atomic structures in these materials and quantifying the atomic displacements, specifically exploring the misorientation effects on measured parameters. The STEM insights are further confirmed by the Raman scattering. Overall, this work sets a high standard for the investigation of similar structures and phenomena.

Reviewer #3 (Remarks to the Author):

The paper has presented a comprehensive study on BST6040 and BaTiO₃ ceramics in nominally paraelectric cubic phase. The size (2-4nm) of the polar nanoclusters in these ceramics was directly observed by the aberration corrected STEM and carefully characterized to calculate the displacement of cations and anion in the unit cell. The observation also reveals that the nature of the polar nano clusters are directly interacting with neighboring nanocluster and embedded in the disordered matrix. The nature of these polar nanoclusters in the BST6040 and BaTiO₃ could help to understand their disordered structure in the materials and resultant of macroscopic property. Some minor correction might be needed.

1, In fig SI1a, the authors put two XRD data in one plot and there should not be actual values showing in the Intensity scale.

2, For materials preparation, it's better to show the raw materials information such as purity and manufacture company. The materials preparation such as calcination temperature and sintering temperature should be provided.

3, The authors might need to refine the XRD data and give crystal details for the studied samples.

4, In line 191 and 192, the heterogeneity in the materials is small scale while homogeneity is on tens of nm in the material. The polar nano clusters are suggested to be 2-4 nm where this scale could be representing the Ba rich region. The polar nano clusters are the property of the BST6040, while the size of them could be associated with the mixing processing.

The comments of each reviewer are **marked in blue font** and each comment is addressed separately. The new text that was added to address these comments is marked by **yellow marker** in the revised manuscript and supplementary information, while the text deleted from the previous version is marked in ~~grey color and strikethrough~~. The exact location of the changes is indicated below.

Reviewer #1

Reviewer #1 comment: This work seems to be done well with a lot of attentions to reliability and accuracy of experimental results. Existence of nano-sized polar region in ferroelectric perovskites is one of the field of debate in current metal oxide functionality. From this viewpoint, this paper seems to be on-time.

Our reply1: We thank reviewer for acknowledging the quality of the experimental results and timing of the manuscript.

Reviewer #1 comment: However, the reviewer wonder if technical importance of this paper is very high to justify publication. This paper focuses on atomic scale disorder. As authors mentions, the averaged chemical composition and properties of (Ba,Sr)TiO₃ solid solution is very homogeneous but locally disorders because of random distribution of Ba and Sr. Indeed, the authors obtained atomic displacement image. However, audiences may not be very surprised with their results because everybody knows distribution of Sr and Ba in (Ba,Sr)TiO₃ should be random and unit cell dimension with Sr and Ba must not be same. There may be some polarization because of the random distribution of Sr and Ba.

Our reply 2: It is possible that we have not been sufficiently clear about the background of our investigation. As we have explained in the paper (see for example pages 2, 8-9 of the original manuscript), the forbidden polarity of dynamic nanosized regions in the cubic, paraelectric phase of BaTiO₃ and KNbO₃ perovskite ferroelectrics has been previously tentatively explained in the literature by dynamic off-center displacements of B-ions (Nb and Ti)^{1,2,3-7}. Origin of these polar nanoregions has not been clarified but has been considered to be due to natural fluctuations in vibrations of central B-site ions in the perovskite ABO₃ unit cell. The theory does not propose that formation of these dynamic polar nanoclusters requires defects, although it can be facilitated by their presence. Importantly, these off-center displacements that have correlation length roughly on the order of about ten unit cells^{1,2} have been considered to be dynamic^{1-3,5}, so that a given orientation of the polarization along one of the eight equivalent pseudocubic <111> directions survives only for 10⁻⁹ to 10⁻¹² seconds.

In addition to directly proving existence of these polar nanoclusters, we show experimentally that, in contrast to the classical theory, the polarization orientation in the polar nanoclusters is stable for hundreds of seconds, that is, the polarization orientation within nanoclusters is practically static. This is a result that goes beyond the reviewer's comment "**there may be some polarization because of the random distribution of Sr and Ba**" and the result is far less straightforward than it could be thought at first hand. Allow us to explain. It is known that addition of Sr to BaTiO₃ helps off-centering of Ti⁸. However, that displacement has been considered to be essential dynamic i.e., Ti jumps among crystallographically equivalent directions with the same energy.^{3,9} We propose that the strain associated with Ba and Sr size difference freezes displacement of Ti atoms along one of these directions resulting in a definite, static polarity of each nanocluster (by the term "static" we mean that the thermal energy is no longer sufficient to flip dipoles among <111> sites, but the dipoles can still be affected by external field, and even flip if the field is large enough^{10,11}). We give evidence of this strain that stabilizes polarization orientation by both STEM and Raman experiments. In addition, we show that all other atoms (A-site, O) are displaced from expected cubic positions, contributing to local

polarization, not only B-site atoms. Therefore, the forbidden polarization within unit cells is probably not oriented only along $\langle 111 \rangle$ directions or any definite crystallographic direction, as previously believed.^{1,2,8} Furthermore, that freezing of polar nanoclusters is not a trivial consequence of mixing ions of different size can be seen by considering $\text{Ba}(\text{Zr}_{0.50}\text{Ti}_{0.50})\text{O}_3$. In this BaTiO_3 -based relaxor material, effective Hamiltonian Monte Carlo simulations with $12 \times 12 \times 12$ supercells^{12,13} showed that polar nano regions are essentially dynamic (orientation of polarization stable at timescales on the order of ps⁹) and are formed in any significant amount only at cryogenic temperatures while in the case of BST and BaTiO_3 we show experimentally that those regions not only occupy large volumes of the sample but are static and stable even above 470 K, allowing them to be investigated by electron microscopy. Zr, which is larger than Ti ($r_{\text{Zr}}=0.86 \text{ \AA}$, $r_{\text{Ti}}=0.745 \text{ \AA}$, $|\Delta r| = 0.115 \text{ \AA}$), does not have the same cell-deforming and polarization-stabilizing character as Sr, which substitutes at the A-site and is smaller than Ba ($r_{\text{Sr}}=1.58 \text{ \AA}$, $r_{\text{Ba}}=1.75 \text{ \AA}$, $|\Delta r| = 0.17 \text{ \AA}$; we use here crystal radii from Shannon¹⁴).

We believe that the presented results change the way in which to look at this decades old problem in physics of ferroelectrics. Based on the currently available literature, the main results presented in this paper are neither obvious nor entirely expected.

The focus of our paper is on breaking of the cubic symmetry on atomic scale in the paraelectric phase of selected ferroelectrics. We show that atomic displacements from ideal cubic positions are such that the resulting nanoclusters are polar. The question of polarity of these stationary regions is nontrivial. Ba and Sr are isovalent, Sr is "non-ferroelectric ion", so polarity of distorted cells is not obvious. Furthermore, in the case of $(\text{Ba,Sr})\text{TiO}_3$ (BST) a random distribution of Ba and Sr is indeed expected on a large scale but is not at all assured on nanoscale by usually used preparation methods. Nanosized pockets with a high concentration of Ba which could lead to ferroelectric clusters at room temperature where STEM studies were conducted cannot be *a priori* excluded. Demonstrating nominal stoichiometry in atomic columns and nanometer scale is technically not trivial and proving nominal Ba/Sr ratio from column to column was essential for the purposes of this paper. By combining modelling (see S14) and STEM we show that the nominal Ba/Sr ratio is maintained in the atomic columns in which we have investigated atomic displacements on nanoscale; that is, there is no concentration of Ba in polar nanoclusters, which could potentially explain the static polar structure.

We have changed Discussion part to directly emphasize these points (see pages 10-11 of the revised manuscript.).

Reviewer #1 comment: However, if we look at usage or application of $(\text{Ba,Sr})\text{TiO}_3$, its averaged bulk properties appears. Hence, it seems that this work is just for visualization of what we can imagine.

Our reply 3: We disagree with the reviewer's assessment that " ...if we look at usage or application of $(\text{Ba,Sr})\text{TiO}_3$, its averaged bulk properties appears. Hence, this work is just for visualization of what we can imagine." We regret if the text might not have been clear and that the referee has missed this important point which we have, however, mentioned at two places in the original manuscript (page 2 and 3). BST is a leading material used as dielectric for filters, tunable dielectrics, capacitors and resonators in GHz frequency range where dielectric losses are of a paramount importance. At microwave frequencies it is small nanoscale objects such as polar nanoclusters described in our work which give the largest contribution to the dielectric loss, one of the most important parameters in these devices¹⁵⁻¹⁷ (Ref. 19-22 in the original manuscript). Therefore, demonstrating existence, determining the size and showing structure and static nature of polarization orientation in these nanoclusters on atomic scale is essential for understanding materials that are indispensable for functioning of modern electronic and communication devices.

We now reinforce this point on page 2 and 11 of the revised manuscript.

Reviewer #1 comment: Regarding disorder in BaTiO_3 , authors ascribed the local inhomogeneity to defects. The conclusion sounds too straightforward.

Our reply 4: In our manuscript we offer a hypothesis that freezing of polarization within polar nanoclusters in the paraelectric phase of BaTiO_3 (purportedly along one of $\langle 111 \rangle$ directions) could be

assisted by defects. See also *Our reply 2* above. We do not say that local polarity is due to defects but that a given direction of polarization may be stabilized by defects. This is not at all a conclusion but a hypothesis, which we do not try to prove and it is not the central point of the paper. The main results of our paper are on BST in which case we are able to prove existence of nanoclusters, determine their size, show that they fill up large areas of the crystal, and that orientation of the polarization is static as opposed to polarization that persists in the same direction only in the nano to picoseconds range. We show that the same type of static polar nanoclusters are present in macroscopically cubic paraelectric phase of BaTiO₃, at 200°C but are absent (i.e., probably become dynamic) at 300°C. The *proposed* role of defects in BaTiO₃ is to freeze direction of polarization within a given nanocluster. Even though in the case of BaTiO₃ this is only a hypothesis it is based on solid grounds. For example, it has been shown that the forbidden second harmonic signal in paraelectric BaTiO₃ is stronger in ceramics than single crystals, and this has been correlated to a larger concentration of defects in the former.⁴ Darlington and Cernik showed a correlation between a tetragonal distortion in paraelectric BaTiO₃ with a small concentration of point defects¹⁸. Wada et al. visualized presence of a disorder-activated Raman mode in the paraelectric phase of pure BaTiO₃ single crystals, and ascribed it to lattice defects.¹⁹ We give evidence for the presence of this disorder-activated Raman mode also in our manuscript. Therefore, our hypothesis (and not a conclusion) does not sound unreasonable or unphysical. We explain this point more on pages 8-10 of the revised manuscript, in a new subsection, *Freezing of polarisation orientation*.

Reviewer #1 comment: Reviewer would like to point a serious difficulty relating to this report. That is, the researchers cannot know the defect concentration in the sample under STEM/TEM observation. As STEM utilizes electron beam with high brightness, there is a chance to induce defects by irradiation during observation. It is, indeed, difficult to quantify defect concentration induced by TEM observation.

Our reply 5: We appreciate that the reviewer has raised the question of stability of investigated materials under electron beam. To address this comment of the reviewer we have made additional experiments on BaTiO₃ and modelled the response of the sample under the electron beam. This analysis is presented as follows.

To test whether a significant number of vacancies in Ba columns could be produced by prolonged irradiation of the sample with electrons, we evaluated intensities of 25 Ba columns from HAADF images acquired at different beam irradiation times (2s, 10s, 20s and 600s). HAADF images were acquired at 68-180 mrad collection semi-angles, with a camera length of 8 cm, spot size 6C (97 pA beam current), i.e. the experimental conditions used in our study. The images presented in the manuscript were typically collected as a stack of 10 individual 512x512 pixel images with 2s frame time (8μs/pixel). In Figure R1 intensities were measured on 1st, 5th and 10th frame of the stack. In the case of 600s the whole area was irradiated under similar conditions (8μs/pixel) for 10 minutes and then the image was taken. We quantified the Ba column intensities from the HAADF images; the intensities of the individual atomic columns in each image were normalized to the highest column intensity in image after 2s of irradiation (assumingly representing column without defects) and the average experimental intensity ratio was calculated as shown in Figure R1. Details of the quantitative HAADF analysis of the atomic column intensities are given in SI4.

From Figure R1, it can be seen that there is no major variation in the average Ba column intensity with time; the variations in the results are within the measurement error. **Therefore, we demonstrate that BaTiO₃ is stable under electron beam at experimental conditions employed in this study.**

Figure R1: Experimental average Ba-column intensity ratios (I_{Ba}) obtained after 2s, 10s, 20s and 600s of electron beam irradiation on BaTiO₃ heated in situ at 473K. See text for details. Bars represent 5% measurement error.

In addition, we irradiated the area with electrons for 10 min at 3.5 times higher beam current than used in our study, i.e., at spot size 3C (357 pA beam current) to demonstrate that the experimental conditions we used are far below those that would potentially damage the sample. Because the column intensity depends strongly on current used, we cannot directly compare the intensities. Therefore, we measured the intensity ratio between Ba and Ti columns for each current used. The average Ba/Ti intensity ratio before and after the experiment was 1.37 +/-0.07 and 1.40 +/-0.07, respectively. In the case that electron beam would induce generation of vacancies we expect that the number of generated vacancies would not be the same for Ba and Ti, i.e. the ratio Ba/Ti should change (reasonably assuming different volatilization tendencies of the two atoms). Our results indicate that the number of vacancies formed in both Ba and Ti atomic columns is negligible even after using harsh TEM conditions.

We can therefore conclude that defects are not produced during measurements under the experimental conditions used in this investigation.

Finally, a third way to test if the beam affects presented results is to test for possible influence of the beam on displacements measurements. If the electron beam induces defects and defects are responsible for displacements of atoms from ideal positions, then displacements should be dependent on the duration of irradiation. To investigate the influence of the electron beam on the displacement measurements, we determine the Ti vs Ba displacements from HAADF images obtained on ~4x4 nm polar cluster in BaTiO₃ in situ heated at 473 K after 2s, 10s and 20s of electron beam irradiation. A total acquisition time of 20 s is the time required to obtain a high-quality STEM image, typically used for displacement analysis. As seen from Figure R2 in all cases, the displacements Ti vs Ba indicate non-cubic symmetry and show similar directions and magnitudes.

We can therefore conclude that the nanoclusters are not formed or modified during observation of the sample in the TEM microscope, but are a property of the material.

Figure R2: Polar figure of Ti vs Ba displacements measured on ~4x4 nm sized polar cluster in BaTiO₃ in situ heated at 473K, after 2s, 10s and 20s of electron beam irradiation. The average measure displacements after 2s, 10s and 20s are 30pm +/-18pm, 35pm +/-18pm, 33pm +/-19 pm, respectively.

We have added results of this analysis in the revised version on page 10 of the revised manuscript and in new Supplementary Information 7 (pages 19-21).

Reviewer #1 comment: It is also difficult to know the defect concentration in the original sample before TEM study. If the authors' sample possesses highly insulating properties, defect concentration in their sample was low enough and likely on the order of 10^{16} cm⁻³ or further less. This number corresponds to approximately one defect among one million atoms. That means defect concentration in BaTiO₃ must be negligibly small compared to Sr fraction at A-site in (Ba,Sr)TiO₃. With this consideration, attributing 'observed' inhomogeneity to defect seems unrealistic. On the other hand, authors' conclusion may be acceptable if defect concentration in BaTiO₃ is a few orders of magnitude higher than reviewer's guess. If this is the case, BaTiO₃ under TEM observation is unlikely similar to ordinary bulk BaTiO₃.

Our reply 6: Before answering reviewer's comment on concentration of defects in BaTiO₃, we address reviewer's comment "That means defect concentration in BaTiO₃ must be negligibly small compared to Sr fraction at A-site in (Ba,Sr)TiO₃". In Ba_{1-x}Sr_xTiO₃, polar nanoclusters are present in the paraelectric phase of all compositions of the solid solution with $x \leq 0.9$ ^{20,21}, including BaTiO₃^{4,22}. We have, for example, observed macroscopic polarization in the paraelectric phase of Ba_{0.975}Sr_{0.025}TiO₃ which has 16 times smaller concentration of Sr than BST6040, which has been studied in detail in this paper. Therefore 40% of Sr in BaTiO₃ is not needed to freeze the direction of polar nanoclusters. It seems plausible to propose that a similar physical effect is responsible for the freezing of polarization within nanoclusters across the phase diagram of Ba_{1-x}Sr_xTiO₃, in all ferroelectric compositions with $x \leq 0.9$, including BaTiO₃. We have performed our study on BST6040 only because in that composition STEM studies could have been made at room temperature.

We come back now to the concentration of defects in BaTiO₃. We can, in fact, estimate the minimum concentration of defects present in the investigated BaTiO₃. For the synthesis of our samples, we have used standard electronic grade powder of BaTiO₃, with 99.95% (metal basis) purity declared by the manufacturer. Regrettably, this information is only indirectly given in SI and Method section of the original paper by quoting an earlier paper²¹. This 0.05% concentration of impurities does not include concentration of vacancies of oxygen, barium and titanium. Vacancies are produced during high temperature processing or to compensate acceptor or donor defects dissolved in the lattice and which are part of the quoted 0.05% impurity concentration. Therefore, the actual concentration of defects is likely on the order of 0.05%, as we now discuss in detail in a new section *Defects* of SI1. This estimate of defects concentration agrees well with the value of dielectric losses, which ranges from <0.01 at 100 kHz to 0.02 at 100 Hz^{20,23} and Curie temperature, T_C (≈ 400 K²⁰ vs. 408 K²¹ in undoped single crystal.), both of which are dependent on defects concentration, as can be seen for example Ref.^{18,24}.

Let us assume that the concentration of defects is 0.05%. For BaTiO₃ concentration of 0.05% (relative to either A- or B-site of the perovskite cell) translates to a concentration of defects of about 7.5×10^{18} cm⁻³ (density of BaTiO₃=6.02 g/cm³; molecular weight=233.2 g/cm³). This is about 5 defects in 10000 host atoms (or 1 in 2000). Actual defects are either isovalent impurities which exert strain on the lattice through ionic size difference and/or aliovalent defects (impurities and vacancies needed for charge compensation) that exert influence both due to ionic size and charge difference. Defects are distributed in three dimensions and not linearly, so in average one defect is separated from a neighbouring defect by about 12-13 unit cells or about 5 nanometers (in all three dimensions). The calculation is based on a cubic unit cell of 0.4 nm and one defect per 12.6x12.6x12.6 cells (≈ 2000 cells). This number agrees very well with the length of regions with coherent polarization in the paraelectric phase of BaTiO₃ proposed by Lambert and Comes¹. In other words, since the polarization within nanoclusters is a correlated effect, freezing of ionic displacements in one unit cell may have an effect on the dipole direction in the neighbouring cells, freezing polarization orientation in the whole nanocluster. Note that even if the defect concentration is ten times lower (1 defect per 20000 host atoms), this would still be about 1 defect per 27 unit cells, or 1 defect every 10 nm, close to the size of polar chains as estimated by Lambert and Comes.

We reiterate that the study of BaTiO₃ is not the central point of our paper. In addition, we do not "attribute 'observed' inhomogeneity to defect[s]". The theory^{1,3,5,6,12,13} predicts that nanosize polar regions form spontaneously in perovskite crystals but the direction of associated polarization is stable only on the range of nanoseconds to picoseconds. Our experiments show that polar nanoregions are frozen and that a given orientation of polarization persist for hundreds of seconds. We propose -do not prove- that the freezing of polarization direction is assisted by defects that are inevitable in real materials or added intentionally.

We ask the reviewer to also consider that we present an independent and strong evidence of polar nanoclusters and associated strain in BaTiO₃ by Raman spectroscopy (Fig. 1 and 4 and associated text), made under completely different experimental conditions than STEM experiments. The samples examined by Raman spectroscopy have not been subjected to electron beam. Those results lead to similar conclusions as STEM investigation.

Considering our replies 5 and 6, and defects consideration in SI1, the BaTiO₃ investigated by STEM is quite similar to ordinary bulk BaTiO₃ used in electronics.

To address these comments of the reviewer on material purity we add new text and experimental results in SI1 (page 3-4) and on pages 9-10 of the revised manuscript.

Reviewer #1 comment: One possible response from the authors should be adding consideration of the defect concentration. The reviewer would like to ask the authors, how much defect concentration has to be assumed to enable them to detect defect related structural relaxation with their techniques. As atomic displacement around point defects have been studied by density functional theory calculations, I suppose the authors can estimate possible STEM images with assumed defect concentration. If they cannot explain observed displacements in BaTiO₃ with reasonable number of defects, that may be an indication that defects are formed during sampling or observation. If the actual defect concentration in BaTiO₃ under STEM observation was much higher than that in ordinary bulk BaTiO₃, conclusion of this paper may be a cause of misleading for defect structure.

Our reply 7: We appreciate reviewer's suggestion to look at first-principles studies (such as DFT) to see what is the number of defects needed to produce the investigated phenomena in BaTiO₃. However, data that would be directly applicable to our study are not available in the literature. Many first principle approaches typically study 2x2x2 supercells (8 unit cells) containing one defect, where the concentration of defects is already 12.5%. It was not possible to find a study where the defect concentration would be on the order of 1% (that would require 100 unit cells) or less, because this would require a supercell size that is difficult to calculate with the computational resources currently available to most research groups. Results of some of available calculations show that, for example, addition of one oxygen vacancy in the unit cell of BaTiO₃ causes a displacement of Ba by 14 pm and displacement of O ions by 6 pm and 23 pm²⁵. In another study, addition of two Fe ions to cubic BaTiO₃ in 2x2x2 super cell, caused a tetragonal distortion of the unit cell by 21 and 30 pm.²⁶ In yet another study with co-doping of BaTiO₃ with Ni and Fe, Ni causes displacement of O atoms by 12 pm, while an oxygen vacancy displaces neighbouring Ti atoms by 28 and 22 pm.²⁷ These displacements predicted by DFT-based computations are in a very good agreement with experimental displacements reported in our study (see Fig. 4). A comment is added in SI3 (page 11).

In BST5050, computations show that addition of Sr causes off-centering of Ti in the range of 12 to 16 pm. These displacements also agree very well to those determined experimentally in our work (see Fig. 2). A comment is added on page 6 of the revised manuscript and in SI2 (page 9).

In our opinion, it is important to have in mind that it is difficult to model effects of defects in the cubic phase of BaTiO₃. The ground state (i.e., fully relaxed state) of BaTiO₃ is rhombohedral ferroelectric. To model cubic phase of BaTiO₃, constraints must be applied on the material. The results then depend on the chosen size of the supercell and applied constraints²⁸. Additional challenge for first-principles studies is that our experiments on BaTiO₃ were made at 200°C and 300°C. For illustration, to compute finite temperature properties of (Ba,Zr)TiO₃ based solid solutions some authors have used effective Hamiltonian Monte Carlo simulations with 12x12x12 supercells^{12,13}. While they have not considered small concentration of defects (i.e., no defects apart from Zr) they did get off-center displacements of Ti by 16 pm, which is again very close to displacements determined

experimentally in our study.

Finally, we simulated column intensities for different concentrations of vacancies, to see which concentration of vacancies may be detected under used experimental conditions. To correlate the normalized intensities of individual Ba-atom columns with the concentration of Ba vacancies inside each column, we compared the experimental intensities with calculated intensities. Calculations were performed using the QSTEM code²⁹ with multislice method and frozen phonon approximation. To include the influence of thermal diffuse scattering (TDS), 30 calculations per one image were used. We created an $Pm-3m$ BaTiO₃ structural model consisting of 8 x 8 x 40 unit cells in [001] zone axis using parameters close to experimental conditions. In Ba columns we introduced 2.5, 5, 10 and 20 at% Ba vacancies. Using these models we then simulated the HAADF images and extracted the corresponding Ba-column intensity ratios between columns with Ba vacancies and those with fully filled Ba positions. From Figure R3 we see at least 20% of Ba vacancies are needed in order to get a statistically relevant result. The method is therefore unsuitable for detection of concentration of Ba-vacancies below ~10%. The experimental average intensity ratio is 0.91 +/- 0.05 which is in the range where no relevant conclusions can be made.

We thus conclude that, as expected, concentration of A-site vacancies in BaTiO₃ columns cannot be determined based on data taken in HAADF experiments. This analysis is added to new SI7 (pages 19-21).

Figure R3: Calculated Ba-column intensity ratio between columns containing Ba vacancies and those with fully filled Ba positions (V_{Ba}/Ba) as a function of Ba-vacancy concentration V_{Ba} (0, 2.5, 5, 10 and 20 at%), and experimental average intensity ratio obtained from five different polar nanoclusters in the BaTiO₃, heated in situ at 473K. Bars represent 5% measurement error defined as relative standard deviation of Ba column intensities without vacancies.

Reviewer #1 comment: I would be happy to review the revised version if further quantitative discussion is given for defect structure and defect concentration in BaTiO₃ in the revised version.

Our reply 7: We have performed additional experiments and quantitative analysis of defects and believe that we have answered all reviewer's comments. In particular, we have shown that the electron beam, under experimental conditions used in our study, does not affect defects structure of the examined materials. Where available in the literature, the computed atomic displacements due to defects are comparable to our experimentally measured displacements. The low concentration of defects in BaTiO₃ corresponds very well to the concentration of defects needed to freeze polarization in nanoclusters with the size observed in our work and predicted theoretically.

Reviewer #2

Reviewer #2 comment: The authors report the experimental observations of polar domains in Sr-doped BaTiO₃ systems in the (macroscopically) cubic phase. The properties and structures of these system have been a subject of intense exploration since the early fifties, yet only recently have the advances in aberration corrected STEM allowed to visualize these structures directly. The authors perform a careful study exploring atomic structures in these materials and quantifying the atomic displacements, specifically exploring the misorientation effects on measured parameters. The STEM insights are further confirmed by the Raman scattering. Overall, this work sets a high standard for the investigation of similar structures and phenomena.

Our reply: We thank reviewer for the positive opinion about our work.

Reviewer#3

Reviewer #3 comment: The paper has presented a comprehensive study on BST6040 and BaTiO₃ ceramics in nominally paraelectric cubic phase. The size (2-4nm) of the polar nanoclusters in these ceramics was directly observed by the aberration corrected STEM and carefully characterized to calculate the displacement of cations and anion in the unit cell. The observation also reveals that the nature of the polar nano clusters are directly interacting with neighboring nanocluster and embedded in the disordered matrix. The nature of these polar nanoclusters in the BST6040 and BaTiO₃ could help to understand their disordered structure in the materials and resultant of macroscopic property. Some minor correction might be needed.

Our reply: We thank reviewer for the positive opinion about our work

Reviewer #3 comment 1: In fig SIIa, the authors put two XRD data in one plot and there should not be actual values showing in the Intensity scale.

Our reply 1: We thank reviewer for pointing out this oversight which has now been corrected

Reviewer #3 comment 2, For materials preparation, it's better to show the raw materials information such as purity and manufacture company. The materials preparation such as calcination temperature and sintering temperature should be provided.

Our reply 2: This information has been already provided by quoting references 1 and 2 of the original SI. However, for the sake of self-consistency, the information is now explicitly reported in SI1 (page 3).

Reviewer #3 comment 3, The authors might need to refine the XRD data and give crystal details for the studied samples.

Our reply 3: We appreciate suggestion of the reviewer. Both (Ba_{1-x}Sr_x)TiO₃ and BaTiO₃ ceramics have been thoroughly analyzed in the literature^{8,30} and it seems to us that this additional effort would be redundant. In a previous work we have prepared compositions of (Ba_{1-x}Sr_x)TiO₃ with x=0, 0.025, 0.33, 0.40, 0.67, 0.90, 0.975 and 1 and XRD reflections in the cubic region and Curie temperature follow Vegard's law,²⁰ attesting to excellent mixture of the elements. As we have pointed out in SI1, we have observed qualitatively similar polar behavior in the paraelectric phase of BST6040 samples prepared in different laboratories, by very different techniques and different starting powders. In BaTiO₃ we observe polarity above T_C both in ceramics and single crystals.²¹ XRD refinement of BST6040 would give some differences in lattice parameters for all these ceramic samples prepared by different procedures. While in principle a study of the average structure for these samples could be interesting, we are not convinced that the serious additional effort that is required for such structural refinement would give us more insight for the present study than what is already available in the literature.

We add in SI1 (page 2) value of the lattice parameter for BST6040 which we determined from presently available data taken on our samples and which agrees very well with the value published by other researchers.³⁰

Reviewer #3 comment 4, In line 191 and 192, the heterogeneity in the materials is small scale while homogeneity is on tens of nm in the material. The polar nano clusters are suggested to be 2-4 nm where this scale could be representing the Ba rich region. The polar nano clusters are the property of the BST6040, while the size of them could be associated with the mixing processing.

Our reply 4: We believe that our analysis presented in SI4 shows that composition of each column is within experimental error equal to the nominal composition. While it is possible that within one column we may have concentrated distribution of Ba atoms in one part of the column, this is unlikely to happen in the majority of the columns within a given polar nanocluster. If the polarity is to be

explained by such concentration of Ba, that trend would then have to repeat for majority of columns in all nanoclusters. Since the intensity signal is dominated by the first 5 nm thick layer (see discussion in SI5), this would mean that there is a layer within the material in which Ba is concentrated over the whole examined region. But then, it would be equally likely that there should be layers where Sr is concentrated and we do not see either of this. To address this issue a comment is added in this sense in SI4 (page 13).

References:

1. Lambert, M. & Comes, R. The chain structure and phase transition of BaTiO₃ and KNbO₃. *Solid State Com.* **7**, 305 (1969).
2. Comes, R., Lambert, M. & Guinier, A. The chain structure of BaTiO₃ and KNbO₃. *Solid State Com.* **6**, 715 (1968).
3. Tai, R. Z. *et al.* Picosecond view of microscopic-scale polarization clusters in paraelectric BaTiO₃. *Phys. Rev. Lett.* **93**, 087601 (2004).
4. Pugachev, A. *et al.* Broken Local Symmetry in Paraelectric BaTiO₃ Proved by Second Harmonic Generation. *Physical Review Letters* **108**, 247601 (2012).
5. Pirc, R. & Blinc, R. Off-center Ti model of barium titanate. *Physical Review B* **70**, 134107 (2004).
6. Zalar, B., Laguta, V. V. & Blinc, R. NMR evidence for the coexistence of order-disorder and displacive components in barium titanate. *Phys. Rev. Lett.* **90**, 037601 (2003).
7. Senn, M. S., Keen, D. A., Lucas, T. C. A., Hriljac, J. A. & Goodwin, A. L. Emergence of Long-Range Order in BaTiO₃ from Local Symmetry-Breaking Distortions. *Phys. Rev. Lett.* **116**, 207602 (2016).
8. Levin, I., Krayzman, V. & Woicik, J. C. Local structure in perovskite (Ba,Sr)TiO₃: Reverse Monte Carlo refinements from multiple measurement techniques. *Phys. Rev. B* **89**, 024106 (2014).
9. Paściak, M., Welberry, T. R., Kulda, J., Leoni, S. & Hlinka, J. Dynamic Displacement Disorder of Cubic BaTiO₃. *Phys. Rev. Lett.* **120**, 167601 (2018).
10. Hashemizadeh, S. & Damjanovic, D. Nonlinear dynamics of polar regions in paraelectric phase of (Ba_{1-x}Sr_x)TiO₃ ceramics. *Applied Physics Letters* **110**, 192905 (2017).
11. Riemer, L. M. *et al.* Macroscopic polarization in the nominally ergodic relaxor state of lead magnesium niobate. *Appl. Phys. Lett.* **117**, 102901 (2020).
12. Akbarzadeh, A. R., Prosandeev, S., Walter, E. J., Al-Barakaty, A. & Bellaiche, L. Finite-Temperature Properties of Ba(Zr,Ti)O₃ Relaxors from First Principles. *Physical Review Letters* **108**, 257601 (2012).
13. Laulhé, C., Pasturel, A., Hippert, F. & Kreisel, J. Random local strain effects in homovalent-substituted relaxor ferroelectrics: A first-principles study of BaTi_{0.74}Zr_{0.26}O₃. *Phys. Rev. B* **82**, 132102 (2010).
14. Shannon, R. D. Revised Effective Ionic Radii and Systematic Studies of Interatomic Distances in Halides and Chalcogenides. *Acta Cryst. A* **32**, 751 (1976).
15. Tagantsev, A. K., Sherman, V. O., Astafiev, K. F., Venkatesh, J. & Setter, N. Ferroelectric materials for microwave tunable applications. *Journal of Electroceramics* **11**, 5–66 (2003).
16. Zhang, H. *et al.* Polar nano-clusters in nominally paraelectric ceramics demonstrating high microwave tunability for wireless communication. *Journal of the European Ceramic Society* **40**, 3996–4003 (2020).
17. Lee, C.-H. *et al.* Exploiting dimensionality and defect mitigation to create tunable microwave dielectrics. *Nature* **502**, 532–536 (2013).
18. Darlington, C. N. W. & Cernik, R. J. The ferroelectric phase transition in pure and lightly doped barium titanate. *J. Phys.: Condensed Matter* **3**, 4555 (1991).
19. Wada, S., Suzuki, T., Osada, M., Kakihana, M. & Noma, T. Change of Macroscopic and Microscopic Symmetry of Barium Titanate Single Crystal around Curie Temperature. *Jpn. J. Appl. Phys.* **37**, 5385–5393 (1998).
20. Biancoli, A. Breaking of the macroscopic centric symmetry in Ba_{1-x}Sr_xTiO₃ ceramics and single crystals. (École Polytechnique Fédérale de Lausanne, Switzerland, 2014). doi:10.5075/epfl-thesis-6366.

21. Biancoli, A., Fancher, C. M., Jones, J. L. & Damjanovic, D. Breaking of macroscopic centric symmetry in paraelectric phases of ferroelectric materials and implications for flexoelectricity. *Nature Materials* **14**, 224–229 (2015).
22. Aktas, O., Carpenter, M. A. & Salje, E. K. H. Polar precursor ordering in BaTiO₃ detected by resonant piezoelectric spectroscopy. *Applied Physics Letters* **103**, 142902 (2013).
23. Hashemizadeh, S. Origins of the macroscopic symmetry breaking in centrosymmetric phases of perovskite oxides. (Ecole polytechnique fédérale de Lausanne, 2017). doi:10.5075/epfl-thesis-8026.
24. Hagemann, H.-J. Loss mechanisms and domain stabilisation in doped BaTiO₃. *J. Phys. C: Solid State Phys.* **11**, 3333–3344 (1978).
25. Sheyla Serrano, Carlos Duque, Paul Medina, & Arvids Stashans. Oxygen-vacancy defects in PbTiO₃ and BaTiO₃ crystals: a quantum chemical study. *Proc. SPIE 5122, Advanced Organic and Inorganic Optical Materials* (2003) doi:10.1117/12.515777.
26. Islam, Md. A., Momin, Md. A. & Nesa, M. Effect of Fe doping on the structural, optical and electronic properties of BaTiO₃: DFT based calculation. *Chinese Journal of Physics* **60**, 731–738 (2019).
27. Maldonado, F., Jácome, S. & Stashans, A. Codoping of Ni and Fe in tetragonal BaTiO₃. *Computational Condensed Matter* **13**, 49–54 (2017).
28. Stashans, A. & Castillo, D. Simulation of iron impurity in BaTiO₃ crystals. *Physica B* **404**, 1571–1575 (2009).
29. Koch, C. Determination of core structure periodicity and point defect density along dislocations. (Arizona State University, 2002).
30. Zhou, L., Vilarinho, P. M. & Baptista, J. L. Dependence of the Structural and Dielectric Properties of Ba_{1-x}Sr_xTiO₃ Ceramic Solid Solutions on Raw Material Processing. *J. Europ. Ceram. Soc.* **19**, 2015 (1999).

REVIEWERS' COMMENTS:

Reviewer #1 (Remarks to the Author):

The authors responded the comments well. It seems that authors disclosed as much as possible results and made sufficient explanation for those materials to support their conclusion. Now, this revised version seems to be accepted for publication.

It is good to publish this paper for activation of discussion in this field. I would like to appreciate authors contribution in this field.

Reviewer #3 (Remarks to the Author):

I am happy with the response and corrections made by the authors. The paper is now recommended for publication.

Response to reviewers' comments on paper NCOMMS-20-41812A "Atomic scale symmetry and polar nanoclusters in the paraelectric phase of ferroelectric materials" by Andreja Bencan, Emad Oveisi, Sina Hashemizadeh, Vignaswaran K. Veerapandiyam, Takuya Hoshina, Tadej Rojac, Marco Deluca, Goran Drazic, Dragan Damjanovic

The comments of each reviewer are **marked in blue font** and each comment is addressed separately.

Reviewer #1 (Remarks to the Author):

The authors responded the comments well. It seems that authors disclosed as much as possible results and made sufficient explanation for those materials to support their conclusion. Now, this revised version seems to be accepted for publication. It is good to publish this paper for activation of discussion in this field. I would like to appreciate authors contribution in this field.

We thank the referee for accepting our paper

Reviewer #3 (Remarks to the Author):

I am happy with the response and corrections made by the authors. The paper is now recommended for publication.

We thank the referee for accepting our paper